# Glycerol Electro-Oxidation in Alkaline Medium with Pt-Fe/C Electrocatalysts Synthesized by the Polyol Method: Increased Selectivity and Activity Provided by Less Expensive Catalysts

**DOI:** 10.3390/nano13071173

**Published:** 2023-03-25

**Authors:** Vanderlei S. Lima, Thiago S. Almeida, Adalgisa R. De Andrade

**Affiliations:** 1Departamento de Química, Faculdade de Filosofia Ciências e Letras de Ribeirão Preto, Universidade de São Paulo, Ribeirão Preto 14040-901, SP, Brazil; 2Departamento de Química, Campus Universitário de Iturama, Universidade Federal do Triângulo Mineiro, Iturama 38280-000, MG, Brazil; thiago.almeida@uftm.edu.br; 3UNESP, National Institute for Alternative Technologies of Detection, Toxicological Evaluation and Removal of Micropollutants and Radioactives (INCT-DATREM), Institute of Chemistry, Araraquara 14800-900, SP, Brazil

**Keywords:** electrocatalysis, glycerol oxidation, Pt-based catalyst, alkaline medium

## Abstract

We have investigated platinum catalysts containing iron as a modifier to obtain catalysts with superior electrocatalytic activity toward glycerol electro-oxidation in an alkaline medium. The electrocatalysts, supported on carbon Vulcan, were synthesized by the polyol method. The physicochemical characterization data showed that the metals were well distributed on the carbon support and had small particle size (2 nm). The Pt:Fe metal ratio differed from the nominal composition, indicating that reducing iron with platinum was difficult, even though some parameters of the synthesis process were changed. Electrochemical analyses revealed that PtFe/C was more active and stable than commercial Pt/C was, and analysis of the electrolysis by-products showed that iron addition to Pt/C boosted the glycerol conversion and selectivity for glyceric acid formation.

## 1. Introduction

Glycerol, a renewable, low-cost, and widely available compound, is a waste product from the transesterification of vegetable oils for biodiesel production—10% m/m of the products is glycerol [1,2,3,4]. Given that the rate of world biodiesel production is high, new applications for glycerol as a platform molecule for the chemical industry must be found [2,3,4,5]. The industrial demand for value-added products, including C3 compounds such as glyceric acid, tartronic acid, glyceraldehyde, dihydroxyacetone, hydroxypyruvic acid, and mesoxalic acid derived from glycerol oxidation, is high because these compounds have numerous commercial applications. They are mainly used in the manufacture of biodegradable polymers and as emulsifiers or raw materials for organic synthesis, so they have high market values [4,5]. In addition, glyceric acid can be used to prepare medicines to treat skin diseases, and it can be employed as an intermediate in serine amino acid synthesis and as an anionic monomer for packaging material, not to mention other medicinal applications [4,5,6].

Direct glycerol electro-oxidation to C3 compounds could become a clean electrosynthesis route. The coupling of this reaction in a fuel cell device allows energy to be partially recovered from the waste fuel, while value-added products are produced [4,5,6,7,8]. Recently, electroreduction processes that produce hydrogen, ammonia, or carbon products have also been investigated as promising ways to add economic value to the glycerol electro-oxidation process [5,7].

On the laboratory scale, platinum-based nanocatalysts are still the most promising anode material for glycerol electro-oxidation [9,10]. Conducting glycerol electro-oxidation in alkaline medium paves the way for reducing the amount of catalyst employed during the process, as the electro-oxidation kinetics of alcohols are faster in an alkaline medium compared to their speed in an acidic medium. Furthermore, non-noble metals are generally more stable in an alkaline medium [7,11,12,13]. Additionally, the presence of a co-catalyst as a second metal added to noble metals, e.g., Pt or Pd, favors bifunctional mechanisms. In these mechanisms, the non-noble metal serves as a source of oxygenated molecules, facilitating fuel oxidation, lowering the reaction overpotential, improving the reaction selectivity, and reducing the energy costs [13,14,15,16,17,18].

Recently, several works have been published using the combination of noble metals (Pt and Pd) with other less noble metals or non-noble such as Bi [19,20], Ru [21,22], Ag [23,24,25], Ni [26,27] in order to lower the cost of catalysts. In this context, glycerol electro-oxidation in the presence of iron as a co-catalyst has been much less frequently investigated.

Iron is the fourth most abundant element in the crust. Platinum and iron easily form alloys with three well-known crystalline structures: one alloy (Pt_x_Fe) and two ordered intermetallic alloys (PtFe and Pt_3_Fe) [28,29,30,31]. Synthesis processes under mild conditions, i.e., without the need for extreme heating or addition of strong reducing agents, cannot provide the desired/nominal chemical composition [29,31,32]. Iron addition to noble metals has been reported to afford more stable and active bimetallic catalysts [32,33,34,35,36]. For instance, γ-Fe_2_O_3_-C addition to Pt/C and PtSn/C improves ethanol electro-oxidation [32,36]. In turn, ultrafine Pt-covered Fe_2_P is a robust electrocatalyst for methanol and ethanol electro-oxidation [37]. A further example is a PtFe alloy supported on multiwalled carbon nanotubes, which displays better activity, stability, and a better anti-CO poisoning ability in methanol electro-oxidation than those of commercial Pt/C [38]. Moreover, a PtFe/C nanocatalyst has proven to be 10 times more active than commercial Pt/C is during ethanol electro-oxidation [39]. Palladium modified with iron has also been reported as an electrocatalyst for the electro-oxidation of different alcohols [33,34,35]. For example, ethylene glycol can be efficiently electro-oxidized in the presence of a durable composite consisting of Fe_2_O_3_, graphitic carbon nitride, and Pd nanoparticles [40].

One of us investigated the differences in the selectivity of glycerol electro-oxidation catalyzed by PdM/C (M = Fe or Mn) [35]. Ex situ (LC-MS) and in situ (FTIR spectroscopy) methods showed that high-value-added products (glycolate, glycerate, and oxalate) were formed with a good yield regardless of the co-catalyst (Fe or Mn). By using DFT calculations, Jin et al. [41] studied how the crystalline structure of PtFe_x_ affects glycerol electro-oxidation and found that a more ordered crystallite structure such as fct (tetragonal centered face) improves the electrocatalytic activity and selectivity for tartronic acid formation as compared to those of the fcc (face-centered cubic) structure. The same research group [42] performed glycerol electrolysis in a basic medium to investigate the PtFe_x_ selectivity as a function of the electrocatalyst crystalline structure and to verify greater selectivity for glyceric acid formation.

The authors of recent publications have attempted to reduce or to eliminate the noble metal content in catalysts by employing non-noble metals. Oliveira et al. [12] added cobalt and iron to nickel and found that the glycerol electro-oxidation potential is too high for application in power source devices, but they verified that applications for electrosynthesis purposes may be interesting. Another publication showed that high-value-added products can be obtained from glycerol in the presence of different compositions of ZnFe_x_Co_2-x_O_4_ spinel oxides [43]. Yung-Jung Hsu and co-workers demonstrate the practical use of Au@NiS_x_ yolk@shell nanostructures for efficient glycerol electro-oxidation (GEOR) to produce tartronic acid, one of the highest value-added intermediates, concomitant with hydrogen evolution reaction (hydrogen fuel) [44].

Here, we have prepared platinum nanoparticles with iron as co-catalyst for glycerol electro-oxidation in an alkaline medium. We have also analyzed the value-added C3 products and the stability of this less expensive electrocatalyst. We know the importance of conductive support in electrocatalyst preparation [45,46]. However, as the objective of the work was to investigate the effect of Fe on Pt catalytic activity, we used Vulcan Carbon as conductive support to prepare all the catalysts due to its low cost, easy acquisition, high surface area, and high conductivity.

## 2. Materials and Methods

### 2.1. Electrocatalyst Synthesis

Pt/C, Fe/C, and Pt_50_Fe_50_/C containing 20% metal loading were prepared by the polyol method [47,48,49]. Briefly, to prepare 200 mg of each electrocatalyst, 100.0 mL of ethylene glycol (J. T. Baker, Mexico City, Mexico) was added to a three-necked flask (250 mL), the pH was adjusted to 13 with NaOH (400 mg), and the flask was subjected to ultrasound for 30 min. After NaOH was completely dissolved, 106.3 mg of H_2_PtCl_6_.6H_2_O (Sigma-Aldrich, Saint Louis, MO, USA) was quantitatively transferred to the flask and subjected to ultrasound stirring for over 30 min. To allow electrocatalyst nanoparticles to form, the mixture was maintained under N_2_ flux and refluxed (130 °C) for 4.5 h. The heating was turned off, the solution was allowed to reach room temperature, and 160.0 mg of thermally activated carbon Vulcan (400 °C under N_2_ atmosphere) was added [50]. The resulting carbon dispersion was placed in the ultrasound bath for 30 min. Finally, the system was magnetically stirred for 24 h to enhance nanoparticle adsorption on Carbon Vulcan. Fe/C and Pt_50_Fe_50_/C were synthesized in a similar way by using 145.0 mg of FeCl_2_.4H_2_O as the iron precursor and 31.9 mg of FeCl_2_.4H_2_O (Sigma-Aldrich, Saint Louis, MO, USA) and 80.0 mg of H_2_PtCl_6_.6H_2_O (Sigma-Aldrich, Saint Louis, MO, USA) as the iron and platinum precursors, respectively. In addition, 200 mg of Pt_50_Fe_50_/C-hydrazine was also synthesized as described above. Briefly, after the iron and platinum salts were dissolved, a stoichiometric amount of hydrazine (N_2_H_4_) was added to the reaction flask to reduce all the metal ions.

The suspensions were filtered through a hydrophobic membrane (0.2 μm PTFE membrane, Millipore, Molsheim, France) and washed with 1.0 L of water (Milli-Q- Molsheim, France) and two 10 mL aliquots of ethanol. The membranes containing the electrocatalysts were dried in an oven at 80 °C for 4 h. After that, the electrocatalysts were removed from the membrane and properly stored.

### 2.2. Physical Characterization

Metal loading in the electrocatalysts was analyzed using simultaneous DTA-TGA equipment, TA Instruments (New Castle, USA), the model SDT 2960 at a constant heating rate of 10 °C min^−1^ from 30 to 900 °C under air atmosphere flow (100 mL min^−1^). Around 5.0 mg of the sample was placed in a Pt-crucible.

The electrocatalyst composition was analyzed by energy dispersive X-ray spectroscopy (EDX) from IXRF System Inc., model 500 Digital Process (Houston, TX, USA) using a scanning electron microscope (SEM), model EVO 50 (Cambridge, UK). 

Diffraction patterns were obtained on an X-ray diffractometer (D5005 from Siemens- Munich, German) operating with Cu Kα radiation (λ = 1.5406 Å) generated at 40 kV and 40 mA. The following parameters were kept constant during the analyses: 2θ range = 30°–90°, step = 0.01° s^−1^, and total analysis time = 100 min. XRD data were corrected by the background and refined by fitting the experimental angular range of interest to the pseudo-Voigt1 function per crystalline peak with a computer refinement program (Profile Plus Executable, Siemens AG, Munich, German). The crystallite size was estimated by using Scherrer’s equation [51]. The electrocatalyst morphology was also investigated by Transmission Electron Microscopy (TEM) using an FEI TECNAI G^2^ F20 HRTEM (200 kV) microscope (Fei Company, Hillsboro, Oregon).

### 2.3. Electrochemical Measurements

The electrode was prepared by the deposition of 5 μL of catalyst ink solution on a previously polished glassy carbon electrode (0.07 cm^2^). The amount of catalyst (m_cat_) that was used to prepare each ink is summarized in the Appendix A. The catalyst ink was obtained by suspending an ideal amount of catalyst (m_cat_) in 100 μL of diluted Nafion^®^ solution (95 μL of isopropanol/5 μL of 5% Nafion^®^) and placing the ink suspension in an ultrasonic bath for 30 min to achieve good dispersion and homogeneity.

The electrochemical profile of the electrocatalysts was obtained by cyclic voltammetry (CV) in an N_2_-purged 0.1 mol L^−1^ NaOH (Mallinckrodt) solution. The measurements were conducted in a conventional electrochemical cell that included Hg/HgO/KOH (0.1 mol L^−1^) and a platinum wire as the reference and counter electrode, respectively. The activity tests were carried out by conducting CV at a scan rate of 10 mV s^−1^, and chronoamperometry (CA) was accomplished in the presence of 0.5 mol L^−1^ glycerol and 0.1 mol L^−1^ NaOH at +0.7 V vs. RHE for 1.5 h. A stability test was performed by recording cyclic voltammograms at 10 mVs^−1^ before and after 1000 CV cycles at 50 mV s^−1^ in support electrolyte (Appendix A). 

The electrochemically active surface area (ECSA) of the electrocatalysts was calculated by the CO stripping technique in 0.1 mol L^−1^ NaOH solution; 0.30 V vs. RHE was applied for 40 min. In the first 15 min, CO was bubbled into the solution and adsorbed onto the electrocatalyst surface. Afterward, the system was purged with N_2_ to remove free CO from the electrolyte solution. The cyclic voltammograms were recorded at a scan rate of 10 mV s^−1^, and ECSA was obtained by integrating the CO voltammetric charge and comparing it with the standard value of 420 μC cm^−2^, which corresponds to the charge required to oxidize a CO monolayer on the platinum catalyst [52,53].

### 2.4. Electrolysis and By-Product Determination

Four-hour-long electrolysis experiments at +0.7 V vs. RHE (−0.4 V vs. Hg/HgO/KOH, 1.0 mol L^−1^) were performed in an electrochemical cell with separate compartments for the cathode and the anode, separated by an anion exchange membrane Fumatech (35 µm thickness). The anode was fabricated by depositing 50 μL of catalyst ink (prepared as described above) over 2.0 cm^2^ (both sides of 1.0 cm × 1.0 cm) Toray^®^ carbon paper.

The reaction products were analyzed by High-Performance Liquid Chromatography (HPLC) on a Shimadzu apparatus equipped with a refractive index (RID) detector. The products were separated on an Aminex HPX-87H column (Bio-Rad Laboratories, Hercules, USA) under isocratic conditions by using 3.33 mmol L^−1^ H_2_SO_4_ at a flow rate of 0.6 mL min^−1^ at 45 °C. Every hour, a 50 μL aliquot of electrolytic solution was injected into the equipment to monitor the formation of by-products.

## 3. Results and Discussion

### 3.1. Electrocatalyst Physical Characterization

Figure 1 presents the thermogravimetric (TG) analysis of the electrocatalysts. Only Pt/C achieved an experimental metal loading value that was equal to the nominal one (20 % wt.). In Fe/C, the amount of iron was 25% higher than that in the nominal composition. The iron content increased probably because part of the iron was converted to its oxide during the TG experiment carried out in an air atmosphere. We also observed that co-reduced iron and platinum in the metal loading decreased significantly compared to the nominal value, indicating that incomplete reduction occurred during the synthesis, or that no metal nanoparticles adsorbed over the carbon support. To clarify this point, we conducted atomic absorption analysis of the solution filtered during the synthesis that employed hydrazine. This analysis indicated that about 70% of the iron remained in the solution. This analysis corroborated the statement above; nevertheless, given that the analytical equipment does not differentiate between Fe^2+^ and Fe^0^, the iron precursor may not have been reduced, or the iron nanoparticles may not have been adsorbed. Considering that hydrazine addition during the synthesis increased the amount of iron in the PtFe electrocatalyst, we can infer that non-reduction might have been the main cause of the lower iron content.

Table 1 summarizes the physical characterization of the electrocatalyst nanoparticles. The EDX analysis of Pt_x_Fe_y_/C evidenced that it was difficult to co-reduce iron with platinum by using the polyol method. The amount of iron obtained at the end of the syntheses was only 5%, rather than the targeted 50%. When we added hydrazine as a reducing agent (Pt_x_Fe_y_/C-hydrazine), the iron content in the composition increased to 25% (half of the desired percentage). However, the use of hydrazine reduced the metal loading over the carbon support to only 6%, as shown by TG analysis. To overcome this low metal loading and iron content, we repeated the synthesis in the presence of hydrazine by changing the parameters such as the pH, amount of hydrazine, reflux time, and temperature, but the results did not improve. Our findings agreed with the literature reports about the difficulty in achieving the nominal composition when dealing with platinum and iron catalysts [28,29,30,31].

Figure 2 illustrates the XRD diffraction patterns and structural profiles of the electrocatalysts. Fe/C did not display any characteristic peak of the cubic structure (space group Im-3m), which should have appeared at 43.50°, 63.20°, and 79.85° (01-071-4650). Therefore, the as-prepared nanoparticles were in an amorphous state. Pt/C exhibited the characteristic peaks of the crystalline structure of face-centered cubic (fcc) platinum. The peaks were related to the reflection planes (111), (200), (220), (311), and (222) (JCPDS # 00-004-0802). The crystallite size remained about 1.4 nm, and the experimental lattice parameter (3.9256 Å) resembled the experimental lattice parameter of pure platinum (3.921 Å).

Pt_95_Fe_05_/C and Pt_75_Fe_25_/C-hydrazine did not present the characteristic peaks of the platinum or the iron structure because low metal loading over the carbon support did not provide enough material for defined diffraction data to be obtained, even at a slow scan rate (0.03° s^−1^). The other reasons for the absence of peaks could be the small crystallite size or the attainment of an amorphous material [54]. For Pt_95_Fe_05_/C and Pt_75_Fe_25_/C-hydrazine, we were able to identify a not well-defined peak at around 40°, which overlapped with the peak at around 46°, both of which are related to the platinum structure. However, the resolution was poor, which prevented the crystallite size or lattice parameter from being calculated.

Figure 3 shows the TEM images and histograms of Pt/C, Pt_95_Fe_05_/C, and Pt_75_Fe_25_/C-hydrazine. The nanostructures were randomly dispersed on the carbon support, and large clusters did not emerge in the materials. The high-resolution TEM images evidenced good crystallinity and a sphere-like shape for the platinum-based electrocatalysts. The particle size remained at around 1.8 nm, and the EDX analysis from TEM characterization agreed well with the results reported in Table 1.

### 3.2. Electrochemical Characterization

Figure 4a depicts the cyclic voltammogram of the platinum-based electrocatalysts in the supporting electrolyte (NaOH 0.5 mol L^−1^), which were recorded from 0.05 to 1.0 V vs. RHE. Peaks in the region between 0.0 and +0.4 V are related to hydrogen adsorption/desorption on the platinum surface [55]. At around +0.80 V (positive sweep) and +0.70 V (negative sweep), the peaks correspond to platinum oxide formation and reduction, respectively [36]. The addition of even small amounts of iron to platinum, as in the case of Pt_95_Fe_05_/C, shifted the hydrogen adsorption/desorption and platinum oxide formation/reduction peaks to less-positive values compared to that of bare platinum. Furthermore, the presence of iron is evidenced in pseudo-capacitive double layer region between 0.4 and 0.6 V vs. RHE by the presence of the Fe^+2^/Fe^+3^ redox couple. Moreover, reversible peaks appeared near 0.6 V vs. RHE (forward scan) and 0.45 V vs. RHE (backward scan), which are associated with iron ion oxidation and reduction and iron oxide formation. Hydrazine addition enhanced the amount of iron in the electrocatalyst (Pt_75_Fe_25_/C-hydrazine), so the redox peaks became more evident. The characteristic peaks of hydrogen adsorption/desorption on platinum sites became poorly defined and the pseudo-capacitive area became wider as an effect of a greater amount of conductive support (Vulcan Carbon) since this composition presented the smallest metal loading value (only 6% wt.), as evidenced by TGA analysis.

The formation of iron oxides at lower potentials is interesting for electrocatalysis: the availability of oxygenated groups facilitates the electro-oxidation of carbonic species on platinum, a consequence of the bifunctional mechanism, during which iron/iron oxide provides oxygenated species for the oxidation process [36,56,57]. As shown in Figure 4b, the electrocatalysts containing iron provided greater activity for glycerol electro-oxidation in an alkaline medium, especially in the case of Pt_75_Fe_25_/C-hydrazine. 

In the presence of 0.5 mol L^−1^ glycerol (Figure 4b), alcohol adsorption on the platinum catalytic sites suppressed the hydrogen adsorption/desorption peaks (+0.05 to +0.4 V vs. RHE). The onset potential for glycerol electro-oxidation was +0.4 vs. RHE for all the investigated electrocatalysts, with very slight variations. An explanation for this is that at high pH values, the supporting electrolyte furnishes OH- species for the alcohol molecule electro-oxidation [10,11,58]. This is a very different situation from acid solutions, where hydroxyl species (-OH) are obtained by water activation on the electrocatalyst surface, and the catalyst site plays a key role in the adsorption and bond cleavage of the water molecules. In a basic solution, the catalytic activity does not depend entirely on this process, so alcohol electro-oxidation starts nearly at the same potential, irrespective of the electrocatalyst composition [59,60].

Shifting toward more positive values, we observed that the oxidation peak was re-activated in the negative scan. This behavior originates from platinum surface regeneration due to the removal/oxidation at higher potentials of adsorbed carbonaceous species produced by direct glycerol electro-oxidation. In the reverse scan, glycerol was electro-oxidized on the renewed surface in a typical alcohol electro-oxidation process [14,55,61,62].

The presence of iron boosted the platinum catalytic activity, as shown in Figure 4b. This behavior depended on the amount of iron. As observed, just 5% wt. of iron (Pt_95_Fe_05_/C) increased the peak current (at +0.87 V vs. RHE) by 35% compared to that of pure platinum. When the iron content was higher, as in the case of Pt_75_Fe_25_/C-hydrazine, the effect was even more significant: the peak current was three times higher compared to that of Pt/C (Table 2).

Our research group has already studied how iron addition affects ethanol and glycerol electro-oxidation. We found that the presence of iron improves the platinum catalytic activity during ethanol and glycerol electro-oxidation in acid and alkaline mediums [35,36,63]. Other authors have also described the use of iron as a co-catalyst and found similar results. The authors claimed that the presence of even small amounts of iron improves the catalytic activity of platinum or palladium, not to mention that iron is much less expensive than commonly used noble metals are, such as ruthenium, palladium, and gold [64,65,66].

Table 3 compares the peak current (I_f_) obtained in the oxidation of glycerol by different materials investigated in the literature. Even considering different experimental conditions, such as concentration of glycerol, support electrolyte and scanning speed, we observed that the PtFe catalyst presents results that are compatible with other compositions that have been investigated. In these cases, the activity is better compared to those ones that use only noble metals (Pt, Pd, and Au) as catalysts.

We carried out chronoamperometry (CA) to investigate the electrochemical performance of electrocatalysts during steady-state glycerol electro-oxidation. We chose the applied potential (+0.7 V vs. ERH) on the basis of the glycerol cyclic voltammograms and because it corresponds to a potential where all the electrocatalysts exhibited electrocatalytic activity toward glycerol electro-oxidation (Figure 4b). Figure 5 shows that the j-t curves were consistent with the cyclic voltammetry curves. The activity of the platinum-based electrocatalysts decreases due to the poisoning of electrocatalytic sites by CO-like species and other glycerol electro-oxidation intermediates, a typical behavior of the electro-oxidation of small molecules on platinum-based catalysts [73].

By analyzing iron addition to platinum in the steady-state studies, we observed that small amounts (5% wt.) of this metal improved the catalytic activity compared to that of bare platinum. This indicates that even a small amount of iron was enough to enhance the reaction kinetics given that its presence helped to renew the platinum catalytic sites. When the amount of iron was 25% wt., the platinum catalytic activity of increased significantly and remained stable. As pointed out earlier, in an alkaline medium, platinum catalytic activity depends on the amount of iron. Proper amounts of iron can act directly on glycerol electro-oxidation through a bifunctional and electronic effect [63].

Figure 6 presents the electrochemical surface area (ECSA) of the electrocatalysts obtained by the CO stripping technique in 0.1 mol L^−1^ NaOH at 10 mV s^−1^ after the CO monolayer was adsorbed. The onset of CO oxidation was at around +0.45 V vs. RHE for Pt/C, and iron addition shifted the CO oxidation to less-positive potentials, 0.3 and 0.40 V vs. RHE for Pt_75_Fe_25_/C and Pt_95_Fe_05_/C, respectively. Moreover, the CO oxidation peak was broader for large potential ranges after iron was added in the electrocatalyst composition. The greater the amount of iron there was, the broader the peak was. This behavior has been reported before and is due to the presence of transition metals on the platinum surface. These metals can modify the platinum electronic structure and contribute to the oxidation of small molecules [16,17,18,74,75].

Figure 6b shows the ESCA measured from the CO stripping experiments and the H_2_ adsorption–desorption region. The data obtained from both methods agree, indicating that both approaches can be used for estimating the electrochemical area.

As shown in Table 2, the ECSA of the PtFe-based electrocatalysts is much higher and remained at around 400 m^2^ g_Pt_^−1^ compared to 240 m^2^ g_Pt_^−1^ for pure platinum. Although the active area is not the only effect that must be evaluated regarding catalyst efficiency, it can significantly contribute to catalytic activity. As seen from the results above, the proper amount of iron and a higher ECSA provide an electrocatalyst with significant activity for glycerol electro-oxidation.

### 3.3. Electrolysis Study

Figure 7 shows the glycerol electro-oxidation by-products and their respective percentages obtained at 4 h of electrolysis (+0.7 V vs. ERH) in the presence of one of the prepared electrocatalysts. In this analysis, we only compared the best iron-containing composition (Pt_75_Fe_25_/C-hydrazine) with Pt/C and the commercial compositions PtRu/C and Pt/C both from BASF. All the investigated compositions electro-oxidized between 5% and 10% of the glycerol present in the solution. In all the cases, the major product was glyceric acid, and the presence of iron or ruthenium in the electrocatalyst improved the selectivity for this by-product, as presented in Figure 7. 

Pt_75_Fe_25_/C provided the best glycerol electro-oxidation as compared to those of the other compositions, including the commercial ones, Pt/C and PtRu/C (BASF) (Figure 7b). Another important feature concerned the effective cost of the electrocatalysts: for instance, Pt_75_Fe_25_/C is much less expensive than Pt/C with Ru as a co-catalyst is. Moreover, iron addition to platinum improved the glycerol electro-oxidation and selectivity for glyceric acid formation.

Figure 8 illustrates the proposed mechanism for glycerol electro-oxidation based on the literature [21,25,62,68]. Considering the products observed by liquid chromatography analysis, the mechanism for PtFe/C follows the solid arrows represented by route II; under the investigated experimental conditions, oxidation occurs via glyceraldehyde formation and not through dihydroxyacetone formation (arrows with dashed lines). In fact, oxidation occurs with the primary carbon to form glyceric acid as the major product. On the other hand, DHA formation (route I) is only observed for Pt/C and PtRu/C. Electrolysis with platinum and ruthenium as electrocatalysts applied to glycerol electro-oxidation also gave glyceric acid as the major product [62] and a mixture of DHA, glyceric acid, and tartronic acid [68], indicating that both routes I and II may occur simultaneously on Pt and PtRu electrocatalysts. 

The selectivity for the by-products of glycerol electro-oxidation in the basic medium depends on the metal associated with platinum or palladium, the applied potential, and the OH concentration, as shown in Table 3. Jin et al. [42] also investigated different Pt_x_Fe_y_/CeO_2_ compositions. They found that glyceric acid was the main product (75%) of glycerol electro-oxidation, which is the same finding as that observed in our study. Fashedemi et al. [34] evaluated FeCo@Fe@Pd/MWCNT-COOH core–shell catalysts in a passive glycerol cell operated until it became inactive. These authors reported that the presence of iron and cobalt contributes to complete glycerol electro-oxidation, with mostly carbonate ions being formed. Zhou et al. [25] studied the PtAg composition in electrolysis experiments at different potentials for 2 h and obtained dihydroxyacetone as the main product. Furthermore, the authors showed that the KOH concentration, the glycerol concentration, and the electrolysis time significantly affected the distribution of the electro-oxidation by-products and the mechanism through which they were formed.

Zhou et al. [26] evaluated how nickel or ruthenium addition affects the platinum catalytic activity and found that both metals favor C3-products and that selectivity depends on the applied potential. González-Cobos et al. [19] evaluated how bismuth addition affects platinum and palladium catalysts and obtained glyceraldehyde as the main product after electrolysis for 4 h.

## 4. Conclusions

The results obtained herein show that addition of iron, an inexpensive metal, is a good alternative for improving the platinum catalytic activity without affecting the process’ efficiency. The main findings of this investigation are: The polyol method provides good metal distribution and small nanoparticles (<2 nm) in PtFe electrocatalysts.Iron addition to platinum increases the active area and the catalytic activity for glycerol and CO electro-oxidation.Initially, glycerol electro-oxidation depends on the type of co-catalyst. PtFe favors the glyceraldehyde oxidation route, and glyceric acid is the main product.PtFe presents high selectivity for high-value-added C3-products.

## Figures and Tables

**Figure 1 nanomaterials-13-01173-f001:**
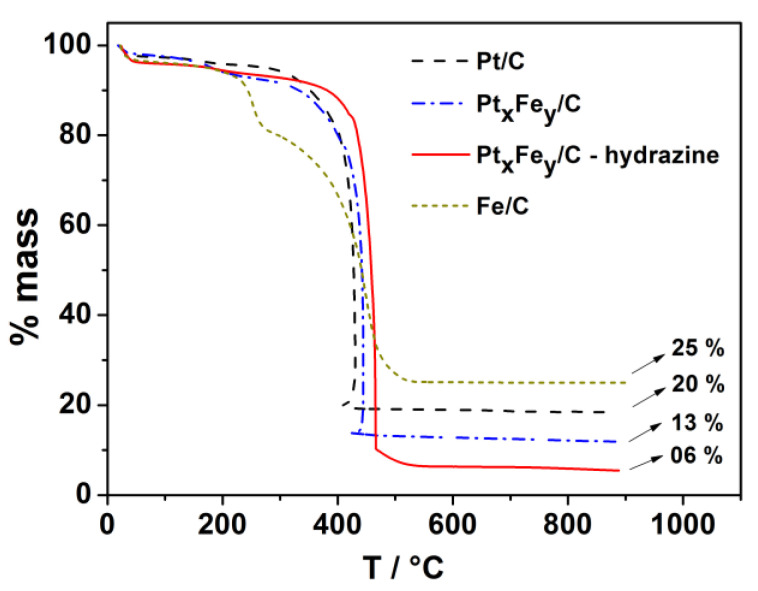
Thermogravimetric analysis curves of the Fe/C, Pt/C, and Pt_x_Fe_y_/C electrocatalysts under compressed air atmosphere recorded at 10 °C min^−1^.

**Figure 2 nanomaterials-13-01173-f002:**
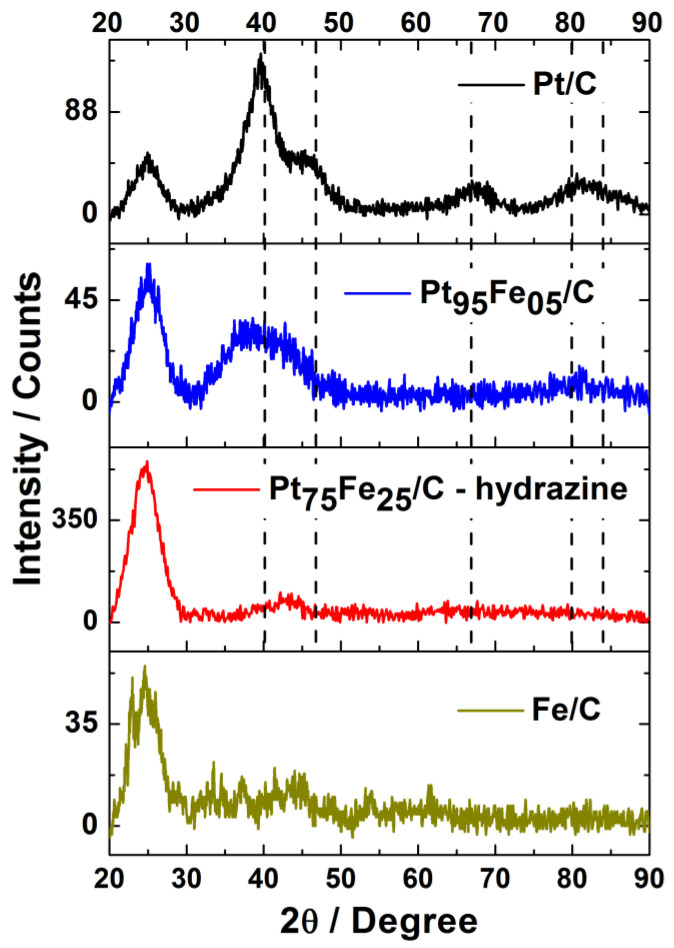
XRD patterns of the Pt and PtFe-based electrocatalysts.

**Figure 3 nanomaterials-13-01173-f003:**
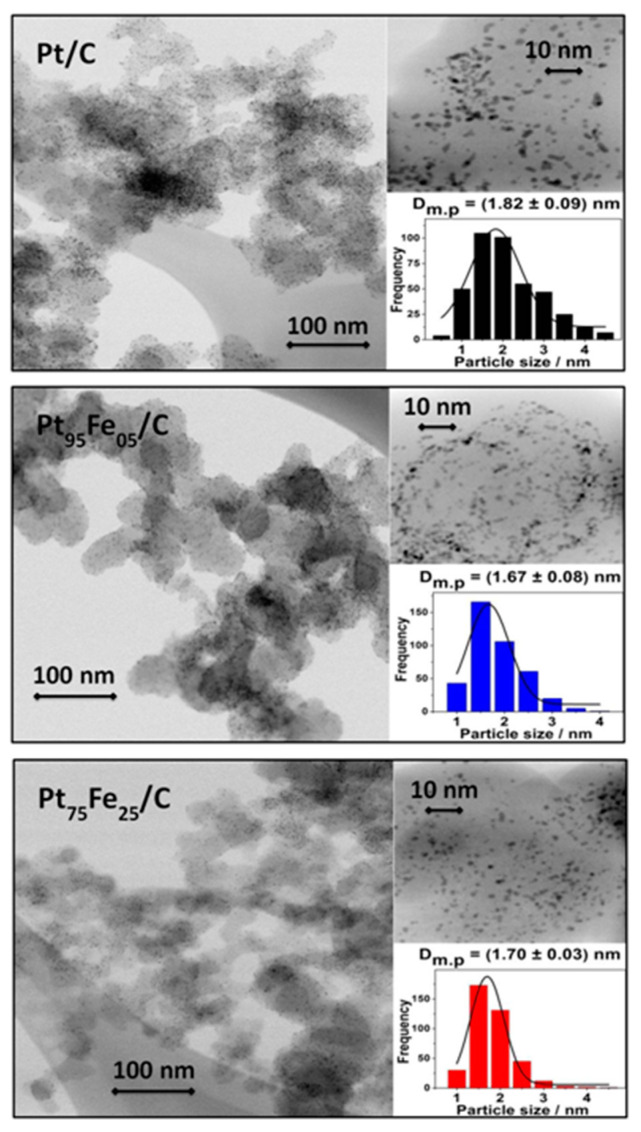
TEM images and histograms of the carbon-supported Pt and PtFe nanomaterials synthesized by the polyol method. Count: 400 particles.

**Figure 4 nanomaterials-13-01173-f004:**
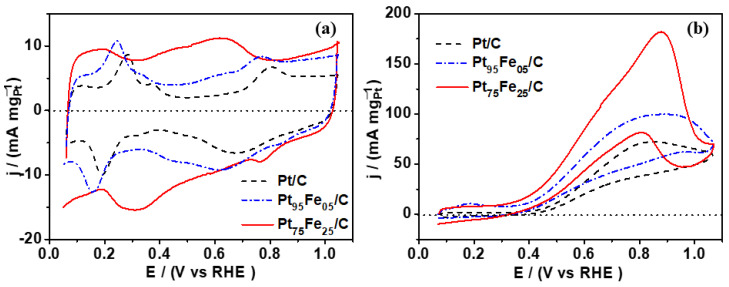
Cyclic voltammogram of platinum-based electrocatalysts in (**a**) NaOH 0.1 mol L^−1^ and (**b**) NaOH 0.1 mol L^−1^ + Glycerol 0.5 mol L^−1^.

**Figure 5 nanomaterials-13-01173-f005:**
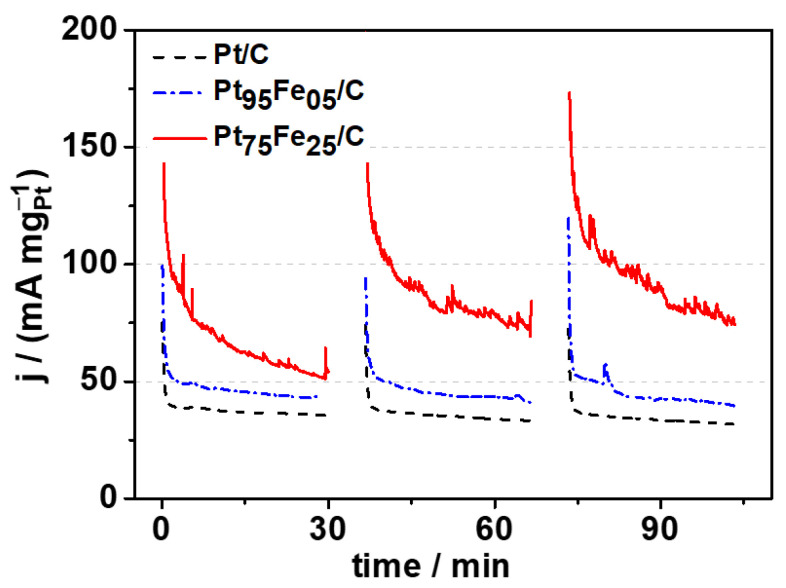
Chronoamperometric curves of the PtFe/C electrocatalyst for glycerol electro-oxidation (0.5 mol L^−1^). Supporting electrolyte NaOH 0.1 mol L^−1^, E_app_ = +0.7 V vs. RHE.

**Figure 6 nanomaterials-13-01173-f006:**
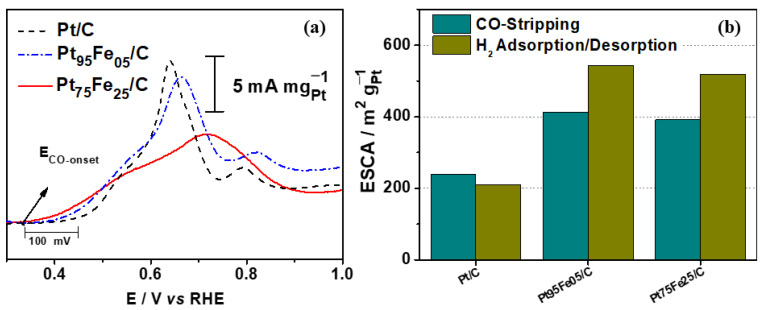
(**a**) CO stripping voltammogram for the platinum-based electrocatalysts in alkaline medium; (**b**) comparison of ECSA obtained by CO stripping and H_2_ adsorption/desorption experiments.

**Figure 7 nanomaterials-13-01173-f007:**
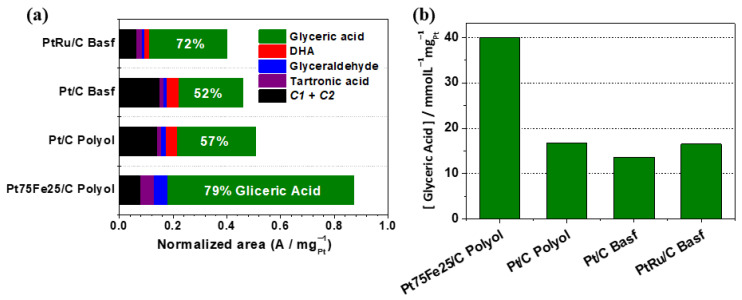
(**a**) Distribution of the glycerol electro-oxidation products as a function of time at +0.7 V vs. RHE at 4 h on the platinum-based electrocatalysts in 0.1 mol L^−1^ NaOH + 0.5 mol L^−1^ glycerol. (**b**) Concentration glyceric acid normalized per mg of platinum.

**Figure 8 nanomaterials-13-01173-f008:**
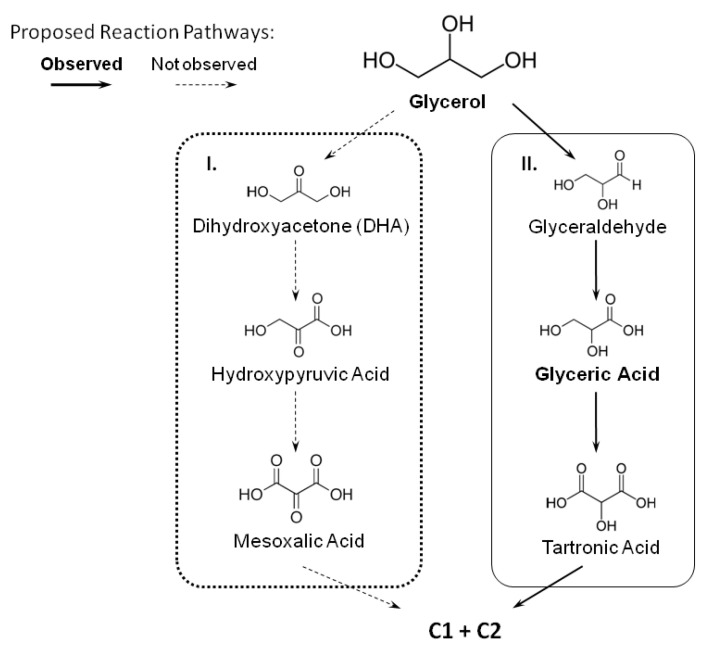
Proposed mechanism for glycerol electro-oxidation vs. that of Pt_75_Fe_25_/C.

**Table 1 nanomaterials-13-01173-t001:** Physical characteristic of the Pt and PtFe electrocatalysts.

Nominal Composition	Experimental Composition (EDX)	TG (%M)	d XRD [nm]	TEM [nm]
Fe/C	Fe/C	25	-	-
Pt/C	Pt/C	20	1.4	1.82
Pt_50_Fe_50_/C	Pt_95_Fe_05_/C	13	-	1.67
Pt_50_Fe_50_/C-hydrazine	Pt_75_Fe_25_/C	6	-	1.70

TG = metal loading, d XRD = crystallite size, and d TEM = particle size.

**Table 2 nanomaterials-13-01173-t002:** Electrochemical activity measurements for platinum-based electrocatalysts.

Composition	Eonset V vs. RHE	Anodic Peak (mA mg_Pt_^−1^) @ +0.87 V vs. RHE	Catalytic Activity (mA mg_Pt_^−1^) @ +0.70 V vs. RHE	ECSA (m^2^ g_Pt_^−1^)
Pt/C	0.40	65	30	240
Pt_95_Fe_05_/C	0.35	88	40	415
Pt_75_Fe_25_/C	0.35	192	75	390

**Table 3 nanomaterials-13-01173-t003:** Comparison of the catalytic activity and selectivity for glycerol oxidation in alkaline media.

Catalyst Composition Synthesis Method	Electrolyte	Peak Current (I_f_) Peak Current Potential (E_f_)	Condition Electrolysis	Selectivity (%) Gly Conversion (%)	[REF] Year
Pt_75_Fe_25_/C Polyol	0.5 M Gly + 0.1 M NaOH	I_f_ = 192 mA mg_Pt_^−1^ (10 mV s^−1^) E_f_ = +0.87 V vs. RHE	E = 0.7 V vs. RHE (4 h)	Glycerate (79%) Glycerol (5–10%)	This Work
PdFe/C Pechini/microwave	0.1 M Gly + 0.1 M NaOH	I_f_ = 28 mA mg_Pd_^−1^ (50 mVs^−1^) E_f_ = 1.2 V vs. RHE	E = 0.8 V vs. RHE	Glycerate (2 mM)	[35] 2014
PdFe/rGO Reduction with NaBH_4_	0.1 M Gly + 1 M KOH	I_f_ = 1.1 A mg_Pd_^−1^ (50 mV s^−1^) E_f_ = 0.9 V vs. RHE	E = 0.8 V vs. RHE (2 h)	Glycerate (45%)	[67] 2021
NiFe/C H_2_ flow at 300 °C	0.1 M Gly + 0.1 M NaOH	I = 60 mA g_M_^−1^ (50 mV s^−1^) E = 1.6 V vs. RHE	E = 1.6 V vs. RHE (8 h)	Formate (4%) Glycerol (14%)	[12] 2014
NiFeCo/C H_2_ flow at 300 °C	0.1 M Gly + 0.1 M NaOH	I = 60 mA g_M_^−1^ (50 mV s^−1^) E = 1.6 V vs. RHE	E = 1.6 V vs. RHE (8 h)	Formate (34%) Glycerol (13%)	[12] 2014
Pt_86_Ru/C Pechini/microwave	0.5 M Gly + 1.0 M NaOH	I_f_ = 525 mA mg_Pt_^−1^ (10 mVs^−1^) E_f_ = 1.2 V vs. RHE	E = 0.7 V vs. RHE (4 h)	DHA (8 mM)	[68] 2017
Ru@Pt/CNTs Reduction with NaBH_4_	0.5 M Gly + 1 m NaOH	I_f_ = 215 mA mg_Pt_^−1^(10 mV s^−1^) E_f_= −0.46 V vs. Hg/HgO	E = −0.2 V vs. Hg/HgO (12 h)	Glycerate Glycerol (40%)	[62] 2017
Ni@Pt/CNTs Reduction with NaBH_4_	0.5 M Gly + 1 m NaOH	I_f_ = 275 mA mg_Pt_^−1^ (10 mV s^−1^) E_f_ = −0.49 V vs. Hg/HgO	E = −0.2 V vs. Hg/HgO (6 h)	Glycerate Glycerol (60%)	[62] 2017
PtAg Galvanic replacement	1 M Gly + 0.1 M KOH	I_f_ = 7.6 mA cm^−2^ (50 mV s^−1^) E_f_ = 1.0 V vs. RHE	E = 0.7 V vs. RHE	DHA (83%)	[25] 2019
PtRh/GNS Polyol	0.5 ML Gly + 0.5 M KOH	I_f_ = 4.5 mA cm^−2^ (50 mV s^−1^) E_f_ = −0.25 V vs. SCE	E = 0.2 V vs. SCE	Glycerate (50%)	[26] 2018
PtNi_2_	0.1 M Gly + 1 M KOH	I_f_ = 2 Amg_Pt_^−1^ (50 mV s^−1^) E_f_ = 0.85 V vs. RHE	E = 0.8 V vs. RHE (2h)	Tartronate (60%)	[69] 2022
Pt@Ag-NaCl Hydrothermal	1 M Gly + 0.1 M KO (I_f_) 0.5 M Gly + 0.5 M KOH (Elect.)	I_f_ = 1.2 mA cm^−2^ (50 mV s^−1^) E_f_ = 1.0 V vs. RHE	E = 0.5V vs. RHE	DHA (75%)	[70] 2020
Pt _95_Bi_05_/TiN-HNWs	0.05 M Gly + 1.0 M KOH (I_f_) 1 M Gly + 1.0 M KOH (Elect.)	I_f_ = 307 mA mg_Pt_^−1^ (20 mV s^−1^) E_f_ = 1.0 V vs. RHE	E = 1.0 V vs. RHE (8h)	Glycerate (37%) Glycerol (87%)	[71] 2022
PtPd@Ag-NH_3_ Hydrothermal	1 M Gly + 0.1 M KO (I_f_) 0.5 M Gly + 0.5 M KOH (Elect.)	I_f_ = 9.2 mA cm^−2^ (50 mV s^−1^) E_f_ = 1.0 V vs. RHE	E = 0.5 V vs. RHE	DHA (70%)	[70] 2020
Pt_4_Au_6_@Ag Hydrothermal	1 M Gly + 0.1 M KOH (I_f_) 0.5 M Gly + 0.5 M KOH (Elect.)	I_f_ = 2.8 mA cm^−2^ (50 mV s^−1^) E_f_ = 1.0 V vs. RHE	E = 1.1 V vs. RHE	DHA (77%)	[72] 2019

## Data Availability

All data included in this study are available upon request upon contacting the corresponding author.

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
