# Peer review of "Glycerol Electro-Oxidation in Alkaline Medium with Pt-Fe/C Electrocatalysts Synthesized by the Polyol Method: Increased Selectivity and Activity Provided by Less Expensive Catalysts"

_nanomaterials, 2023, doi:10.3390/nano13071173_

Round 1
Reviewer 1 Report
The manuscript entitled “Glycerol Electrooxidation in alkaline medium with Pt-Fe/C electrocatalysts synthesized by the Polyol Method: Increased Selectivity and Activity Provided by Less Expensive Catalysts” consists of the study of the modification of Pt-based electrocatalysts for the glycerol electrooxidation. The manuscript provides some useful information, but I found some points that must be addressed before publishing the manuscript.
1. The carbon support has proven to be of paramount importance in electrocatalysis for Pt/C materials. Here only Vulcan has been provided and the textural properties of the materials has not been studied. There are many articles in literature showing the relevance of the carbon support. The authors need to justify the selection of the Vulcan material. I recommend the authors to include the following literature: DOI: 10.1016/j.jpowsour.2004.06.075 ; DOI: 10.3390/c7020032.
2. TGA is not the most accurate technique to provide metal loading in materials science. I recommend performing ICP experiments. Likewise, the authors provided the Pt content from EDX, which is again not recommendable. ICP (or XPS) can give more information about the chemical composition.
3. The authors mentioned that the higher current density observed in the CV is related to the pseudo-capacitance of Fe. How is that possible? Fe is a metal that only shows redox process related to the modification of the oxidation states. Somehow, Fe is modifying the textural properties of the material or increasing the wettability of the electrocatalysts, which reinforce point 1 argument.
4. Figure 4b shows the CV cycles for Pt/C, Pt95Fe05/C and Pt75Fe25/C, but not for the Fe/C electrocatalysts. Fe has proven to efficiently electrocatalyse many reaction, such as the ORR, CO2, etc. The authors need to include the Fe/C and explain the different behaviour of Fe compared to the other electrochemical reaction. Here some recent references that I recommend to include during the discussion: DOI: 10.1016/j.cej.2022.140858; DOI: 10.1002/adfm.202300405; 10.1002/eem2.12278
Author Response
We would like to thank the reviewer for his/her insightful comments on the paper, as these comments led us to improve the final version of the manuscript. Our revision reflects all suggestions. Detailed responses are given below.
Reviwer #1
- The carbon support has proven to be of paramount importance in electrocatalysis for Pt/C materials. Here only Vulcan has been provided and the textural properties of the materials has not been studied. There are many articles in literature showing the relevance of the carbon support. The authors need to justify the selection of the Vulcan material. I recommend the authors to include the following literature: DOI: 10.1016/j.jpowsour.2004.06.075 ; DOI: 10.3390/c7020032.
Thanks for the observation. We understand the importance of conductive support in electrocatalysts preparation, however, as the focus of this work was to evaluate the contribution of Fe in the catalytic activity of Pt towards Glycerol oxidation, we thought that it would be more convenient to use well-known carbon support whose properties of high surface area and conductivities are already reported in the literature, as well as its low costs and easy acquisition. We are making this note in the manuscript and including the recommended references. (Line 98-102)
- TGA is not the most accurate technique to provide metal loading in materials science. I recommend performing ICP experiments. Likewise, the authors provided the Pt content from EDX, which is again not recommendable. ICP (or XPS) can give more information about the chemical composition.
We agree that TGA and EDX only provide a semi-quantitative analysis of the catalyst composition, and we have introduced this comment in the manuscript. However, due to the difficulty in catalyst synthesis, the amount of catalysts was less than expected and was sufficient to carry out ICP analysis.
- The authors mentioned that the higher current density observed in the CV is related to the pseudo-capacitance of Fe. How is that possible? Fe is a metal that only shows redox process related to the modification of the oxidation states. Somehow, Fe is modifying the textural properties of the material or increasing the wettability of the electrocatalysts, which reinforce point 1 argument.
We agree with the remarks made and change this comment in the manuscript. (lines 291-303)
The difference in the pseudo-capacitive region is due to the effect of the amount of Carbon Vulcan (conducting support) in the compositions. As shown by the TGA curves, the compositions containing Fe are those with the highest amount of Carbon support and therefore a change in the capacitive charge of the voltammogram is to be expected. In fact, the presence of iron is evidenced by the appearance of redox peaks/regions only. We will make the correction in the manuscript.
Figure: Cyclic voltammogram (10 mVs-1) in 0.1 mol L-1 NaOH with pure glassy carbon (C) electrode and glassy carbon with Vulcan Carbon and Fe/C with a mass of 0.10 mg.
- Figure 4b shows the CV cycles for Pt/C, Pt95Fe05/C and Pt75Fe25/C, but not for the Fe/C electrocatalysts. Fe has proven to efficiently electrocatalyse many reactions, such as the ORR, CO2, etc. The authors need to include the Fe/C and explain the different behavior of Fe compared to the other electrochemical reaction. Here some recent references that I recommend to include during the discussion: DOI: 10.1016/j.cej.2022.140858; DOI: 10.1002/adfm.202300405; 10.1002/eem2.12278
Figure 4 we normalize the current by the amount of noble metal introduced (Pt). Although Fe/C has been used with success as a cathode as can be shown in the figure above the C/Fe shows a low activity when applied to the anode side and does not show any oxidation peak of glycerol.

Reviewer 2 Report
This work reported the use of Pt-Fe/C electrocatalysts for electrochemical oxidation of glycerol. The topic of the work was significant with regard to the development of electrocatalysis technology. The manuscript can be considered for publication after the following issues are addressed.
Comments:
(1) What if the content of Fe was further increased? Will the activity increase accordingly?
(2) In Fig 5C, the chemical states of the catalysts repeated operation should be examined.
(3) A global activity comparison with the-state-of-the-art electrolysts ever reported for glycerol electrooxidation should be provided in order to highlight the merits of the current system.
(4) Recent development on the use of novel electrocatalysts for similar glycerol electrooxidation should be briefly introduced and cited to enlighten the readers: Adv. Funct. Mater. 2023, vol.33, pp.2209386.
Author Response
We would like to thank the reviewer for his/her insightful comments on the paper, as these comments led us to improve the final version of the manuscript. Our revision reflects all suggestions. Detailed responses are given below.
Reviewer # 02
(1) What if the content of Fe was further increased? Will the activity increase accordingly?
This is a good question that we will keep in mind to be investigated in future studies. Unfortunately, under the milder conditions investigated the difficulty of reducing Fe together with Pt did not allow us to evaluate systematically the effect of Fe amount on the catalytic activity as we wished. We have investigated Pd-Fe catalysts (doi.org/10.1016/j.electacta.2018.10.187) and in the presence of Pd (which did not show reducing issues) increasing the amount of Fe significantly increases the catalytic activity of the composition for ethanol oxidation in alkaline medium So, we expect the Pt-Fe catalysts to show the same behavior.
(2) In Fig 5C, the chemical states of the catalysts repeated operation should be examined.
We apologize, but we don't understand the question as we do not have a figure named 5C.
(3) A global activity comparison with the-state-of-the-art electrolytes ever reported for glycerol electrooxidation should be provided in order to highlight the merits of the current system.
We thank the reviewer for the suggestion and we organized a table in the manuscript to compare the catalytic activity and selectivity of our composition with other studies in the literature. (line 366)
(4) Recent development on the use of novel electrocatalysts for similar glycerol electrooxidation should be briefly introduced and cited to enlighten the readers: Adv. Funct. Mater. 2023, vol.33, pp.2209386.
We thank for the suggestion. we inserted the reference. (ref. 44)

Reviewer 3 Report
This paper describes the Glycerol Electrooxidation in Alkaline Medium with Pt-Fe/C Electrocatalysts Synthesized by the Polyol Method. After reviewing it, I think it can be considered to publish if the following issues are addressed:
1. Some important data is missing, for a kind of nanomaterials, the xps data is necessary for the chemical environment test. Especially for the materials stability test.
2. For the TEM data, the elemental mapping image of the asprepared samples should be added.
3. As the molar ratio between Pt and Fe is the key factor for this work. More data to confirm this should be providedk, such like the ICP-OES/MS.
4. The selectivity of the reaction should be compared with other work. A comprehensive table should be added and discuss in details.
5. For the introduction part, the most recent achievements about this area should be clarified and how necessary the authors investigate should be discussed in detail.
Author Response
Manuscript ID: nanomaterials-2272596
Type of manuscript: Article
Title: Glycerol Electrooxidation in Alkaline Medium with Pt-Fe/C
Electrocatalysts Synthesized by the Polyol Method: Increased Selectivity and
Activity Provided by Less Expensive Catalysts
Authors: Vanderlei Silva Lima, Thiago Santos Almeida, Adalgisa Rodrigues De
Andrade *
We would like to thank the reviewer for his/her insightful comments on the paper, as these comments led us to improve the final version of the manuscript. Our revision reflects all suggestions. Detailed responses are given below.
Reviewer 03
- Some important data is missing, for a kind of nanomaterials, the xps data is necessary for the chemical environment test. especially for the materials stability test.
We were not able to make the XPS of this material before and after the electrochemical analysis to follow the stability, we understand that XPS would not be the best technique to follow the stability of the material. The electrochemical test provides a good insight into the stability of this material. We make 1000 CVs of the electrocatalyst and observed that the introduction of iron improved the stability. This test has been introduced in the supplementary material (Fig. S1).
- For the TEM data, the elemental mapping image of the as prepared samples should be added.
We are very sorry but we did not include this analysis in the TEM analysis. This is a good question that we will keep in mind to be investigated in future studies. Unfortunately, we do not have access to TEM analysis so quickly.
- As the molar ratio between Pt and Fe is the key factor for this work. More data to confirm this should be provided, such like the ICP-OES/MS.
We agree that TGA and EDX only provide a semi-quantitative analysis of the catalyst composition, we have introduced this comment in the manuscript. However, due to the difficulty in catalysts synthesis, the amount of catalyst was less than expected and was sufficient to carry out ICP analysis.
- The selectivity of the reaction should be compared with other work. A comprehensive table should be added and discuss in details.
We thank for your suggestion and we organized a table (line 336) to compare the catalytic activity and selectivity of our composition with other studies in the literature. In addition, we insert below a figure that relates the selectivity for the major products obtained in electrolysis experiments using different materials. Our best composition is highlighted in red.
- For the introduction part, the most recent achievements about this area should be clarified and how necessary the authors investigate should be discussed in detail.
We thank for the suggestion we change introduction.

Round 2
Reviewer 1 Report
The authors have adressed my comments. Therefore, I recommend the publication of this manuscript.
Author Response
We thank again the review to the comments provided.
Reviewer 2 Report
The revised manuscript is now in a good shape. Only an issue was left. For the previous question (2), in Fig 5, the chemical states of the catalysts after repeated operation should be examined by XPS.
Author Response
Unfortunately, we were not able to make the XPS of this material before or after the electrochemical analysis to follow the stability, although the chemical state of the material is interesting this was not the goal of this manuscript at this stage, we understand that XPS would not be the best technique to follow the stability of the material. The electrochemical test provides a good insight into the stability of this material. We make 1000 CVs of the electrocatalyst and observed that the introduction of iron improved the stability. This test has been introduced in the supplementary material (Fig. S1).